# Cafeteria and Fast-Food Diets Induce Neuroinflammation, Social Deficits, but a Different Cardiometabolic Phenotype

**DOI:** 10.3390/nu17223614

**Published:** 2025-11-19

**Authors:** Andrej Feješ, Petronela Sušienková, Lucia Mihalovičová, Veronika Kunšteková, Radana Gurecká, Veronika Borbélyová, Peter Celec, Katarína Šebeková

**Affiliations:** 1Institute of Molecular Biomedicine, Faculty of Medicine, Comenius University Bratislava, Sasinkova 4, 811 08 Bratislava, Slovakia; andrej.fejes@fmed.uniba.sk (A.F.); susienkova8@uniba.sk (P.S.); lucia.mihalovic@gmail.com (L.M.); v.sarayova@gmail.com (V.K.); veronika.borbelyova@fmed.uniba.sk (V.B.); peter.celec@fmed.uniba.sk (P.C.); 2Institute of Epidemiology and Prevention, Faculty of Public Health, Slovak Medical University, Limbová 12, 83303 Bratislava, Slovakia; 3Institute of Medical Physics, Biophysics, Informatics and Telemedicine, Faculty of Medicine, Comenius University Bratislava, Sasinkova 2, 811 08 Bratislava, Slovakia; radana.gurecka@fmed.uniba.sk

**Keywords:** metabolic syndrome, obesity, carbonyl stress, GFAP, 5-HIAA, continuous metabolic syndrome score, indoxyl-sulphate, D-lactate, sociability

## Abstract

Background: Obesity is a risk factor for several non-communicable diseases and premature death. The Western-type diet, rich in calories and diverse in tastes, smells, and textures, promotes the onset and progression of obesity. We compared the effects of two Western-style palatable obesogenic diets—the cafeteria (CAF) diet, which allows for self-selection of calorie-dense food items consumed by humans, and the fast-food diet (FFD)—composed of a fixed combination of cheeseburgers and fries—on the manifestation of obesity-related complications. Methods: 3-month-old female rats consumed either the control (CTRL), FFD, or CAF diet for 12 months. Body weight was monitored weekly. At the end of the experiment, rats underwent metabolic and behavioral testing. Cardiometabolic markers and those characterizing glycoxidative and carbonyl stress, inflammatory status, and tryptophan metabolism were determined. Results: The CAF rats gain most weight (CTRL: +111 ± 40 g; FFD: +211 ± 77 g; CAF: 316 ± 87 g). CAF feeding produced a classical metabolic syndrome–like profile with severe obesity, insulin resistance, dyslipidemia, and liver steatosis, whereas the FFD model led to moderate obesity with preserved insulin sensitivity but elevated blood pressure and hepatic cholesterol accumulation. Thus, the CAF group developed a severe metabolic syndrome-like pathology assessed as continuous metabolic syndrome z-core (CTRL: −2.3 ± 1.0; FFD: −0.4 ± 1.9; CAF: 3.0 ± 2.4). Despite these differences, both diets promoted neuroinflammation and social deficits, likely mediated through gut microbiota–derived metabolites such as 5-HIAA and indoxyl sulfate. Conclusions: In female rats, self-selected CAF diet drives more severe and distinct pattern of metabolic syndrome-like pathology than a fixed FFD.

## 1. Introduction

Obesity is a chronic, multifactorial disease characterized by excessive fat accumulation that increases the risk of chronic non-communicable diseases and premature death. Over the past few decades, obesity has become a global pandemic, affecting more than 30% of the population [1]. Although genetic and environmental factors play a role, consuming a Western-style diet—high in fats, sugars, and salt—significantly increases the incidence of obesity. Disbalance between energy intake and expenditure promotes fat accumulation in adipose tissue and peripheral organs. In obesity, adipose tissue undergoes morphological and endocrine remodeling, triggering low-grade inflammation (including neutrophil activation, macrophage polarization towards a pro-inflammatory phenotype, and the production of pro-inflammatory cytokines), insulin resistance, and organ dysfunction [2]. Therefore, understanding the impact and consequences of dietary composition on metabolic status is crucial for elucidating the pathomechanism of diet-induced obesity (DIO) and for implementing effective measures to prevent obesity development.

Obesity was once attributed mainly to an imbalance between energy intake and expenditure, but its rising prevalence suggests that more complex mechanisms are at play [1,3]. Beyond adipose tissue expansion, obesity involves dysregulated hypothalamic circuits that govern hunger and satiety [4]. Anticipatory insulin signaling, coupled with postprandial hyperglycemia, can lead to hyperinsulinemia that paradoxically drives hunger and overeating [5]. Excess adipose tissue releases pro-inflammatory cytokines that penetrate the blood–brain barrier, damaging microglia and astrocytes [6]. Furthermore, overeating itself promotes neuroinflammation, vascular pathology, and neuronal loss [7]. These metabolic and inflammatory disturbances reinforce one another, creating a vicious cycle that contributes to neurodegeneration.

In rodent models, obesity is commonly induced by high-fat (HFD, typically based on lard or vegetable oil) or high-fat/high-sugar diets. While effective, these diets poorly replicate the sensory complexity, palatability, and voluntary food selection characteristic of human eating behavior [8]. Commercial Western-style chows enriched with lard and sugar partially mimic human diet composition but lack sensory stimulation and choice. To address this, the cafeteria diet (CAF) was developed, offering diverse, palatable human foods with varied tastes, smells, and textures, and later standardized by Lalanza and Snoeren [9]. In rodents, the obesogenic effects of a CAF diet have been compared only with those of an HFD [10,11,12]. In rats fed aroma-containing feed, neuronal activity in brain regions involved in the central regulation of food intake and energy balance is increased, and volatile aroma compounds in foods likely influence endocrine signaling and neuronal regulation of metabolism [13]. Moreover, FFD consumption alters taste, further affecting food intake, creating a vicious cycle of overeating and taste impairment [14]. Here, we compare two palatable obesogenic diets with attractive taste and smell—the CAF diet, which enables voluntary self-selection, and a fast-food diet (FFD), consisting of fixed palatable items—on obesity-related metabolic and behavioral outcomes. We hypothesized that prolonged voluntary selection in the CAF diet would exacerbate cardiometabolic and behavioral disturbances more strongly than the fixed-item FFD.

## 2. Materials and Methods

### 2.1. Animals

Three-month-old female Wistar rats (n = 29) obtained from Velaz (Prague, Czech Republic) were housed 2/cage under a controlled environment (temperature: 24 ± 2 °C, humidity: 55 ± 10%, 12-h light/dark cycle) with ad libitum access to food and water. All experimental procedures were conducted following the EU Directive 2010/63/EU and ARRIVE 2.0 Guidelines [15], after the approval of the protocol by the Ethics Committee of the Institute of Pathophysiology, Medical Faculty, Comenius University, Bratislava (07/2023), and the State Veterinary and Food Administration of the Slovak Republic (No. 4503-3/2024-220).

### 2.2. Experimental Procedures

After 2 weeks of acclimatization, rats were divided into 3 experimental groups of similar body weight and maintained for 12 months on either a standard chow diet (CTRL; n = 10, Catalog No. E15051-04; Ssniff Spezialdiäten GmbH, Soest, Germany) or one of two palatable obesogenic diets. A fast-food diet (FFD; n = 10) consisted of cheeseburgers (composed of a wheat bun, beef patty, processed cheese, pickles (cucumber), mustard, ketchup, and onion) and French fries, which were homogenized by grinding, aliquoted, and stored at −20 °C [14]. The cafeteria diet (CAF; n = 9) consisted of 16 different salty, sweet, and processed food items of varying textures, served in four menus that changed over two days, along with a standard chow [9]. The composition of the fast food (FF) and CAF diet is given in Appendix A, respectively. Food consumption was measured in three 3-week periods during the 3rd and 6th months of the study, and again at the end, before behavioral and metabolic testing had been initiated. Caloric intake was calculated. Body weight was recorded weekly.

Below, the methods are listed in the order in which they were performed on rats according to the protocol.

### 2.3. Oral Glucose Tolerance Test

After 12 h of food deprivation, blood was collected from the tail-end snip to measure the fasting plasma glucose (FPG, Accu-Chek Performa glucometer, Roche Slovakia s.r.o., Bratislava, Slovakia) and insulin (FPI, Rat Insulin ELISA, Mercodia, Uppsala, Sweden). Thereafter, animals were orally gavaged with a glucose solution (2 g/kg; D-(+)-Glucose H_2_O p.a., Centralchem s.r.o., Bratislava, Slovakia). Glycemia and insulinemia were measured at 15, 30, 60, 90, and 120 min after the glucose load. The quantitative insulin sensitivity check index (QUICKI) was calculated: *QUICKI* = 1/[*log*(*FPI,* µIU/mL) + *log*(*FPG*, mg/dL)] [16]. The area under the curve (AUC) of glucose or insulin was calculated. Hepatic and muscular insulin resistance indexes were calculated [17].

### 2.4. Blood Pressure Measurements

Blood pressure was measured noninvasively using the tail-cuff method (CODA^®^ Monitor, Kent Scientific Corporation, Torrington, CT, USA) under mild short (5 min) inhalation anesthesia (97% oxygen, 3% Isoflurane, Vetpharma Animal Health, Barcelona, Spain). The average of three consecutive readings obtained during the last 2 min was considered the final systolic blood pressure (SBP) and diastolic blood pressure (DBP).

### 2.5. Body Composition Analysis

Bone mineral density (BMD), bone mineral content (BMC), lean and abdominal fat mass (analyzed manually after delineation the region between L3-S3 vertebrae) were determined using dual-energy X-ray absorptiometry using the Horizon DXA System (Hologic, Marlborough, MA, USA) equipped with software for small animals (Hologic APEX Software version 5.6.0.5, Bredford, MA, USA) in rats anesthetized by i.p. injection of ketamine (100 mg/kg, Richter Pharma AG, Welse, Austria) and xylazine (10 mg/kg, Ecuphar N.V., Oostkamp, Belgium). The device was calibrated using the Small Animal Step Phantom. The mean of three consecutive measurements per animal was calculated.

### 2.6. Body Temperature Measurements

Rectal body temperature was measured with a thermometer (TFA Dostmann GmbH & Co. KG; Wertheim-Reicholzheim, Wertheim, Germany). An average of three measurements per rat was recorded.

### 2.7. Urine Analyses

To collect stool-free urine, rats were placed into metabolic cages for 24 h with free access to water. Diuresis was recorded, and osmolality was measured using a K-7400S Semi-Micro Osmometer (KNAUER, Berlin, Germany). Aliquoted urine samples were stored at −80 °C. Creatinine was measured using the Jaffe method. Creatine clearance was calculated and normalized to the kidney’s weight. Spectrophotometric methods were used to determine proteinuria (pyrogallol red method) and D-lactate (D-Lactate Colorimetric Assay, Sigma-Aldrich, St. Louis, MO, USA). Neutrophil gelatinase-associated lipocalin (NGAL, Crystal Chem; Elk Grove Village, IL, USA), 5-hydroxy-3-indoleacetic acid (5-HIAA, LDN, Nordhorn, Germany), and indoxyl sulphate (IS; Antibodies, Europe AB, Stockholm, Sweden) were analyzed by ELISA methods according to the manufacturer’s instructions, using the Sapphire II instrument (Tecan, Vienna, Austria). Renal excretion of analytes is given for 24 h.

### 2.8. Behavioral Testing

Before behavioral testing, animals were habituated to the testing room environment (25 ± 2 °C, 55 ± 10% humidity) for 30 min. All tests were recorded and analyzed using automatic tracking software (EthoVision XT 10.0; Noldus Information Technology, Wageningen, The Netherlands).

#### 2.8.1. Open Field Test

The animal was allowed to explore the arena (1 m^2^) freely for 10 min. The locomotor activity was evaluated as the cumulative distance traveled (cm). Time spent in the center zone (s) was recorded as anxiolytic-like behavior.

#### 2.8.2. Elevated Plus Maze Test

The plus maze was situated 45 cm above the ground and consisted of two closed (dark) and two open arms, each measuring 15 cm × 100 cm. The open arms were illuminated at 20 Lux. The animal was placed at the entrance to the open arm and allowed to explore the arena for 5 min. Time spent in the open arms was recorded and considered anxiolytic-like behavior.

#### 2.8.3. Novel Object Recognition Test

To assess short-term memory, the novel object recognition test was used. During the habituation period, the animal was placed into the testing arena (1 m^2^) with two identical objects (plastic cylinders placed in the opposite corners) for 10 min. After a 1-h retention interval, one of the objects (alternating sides with each trial) was replaced by a novel object—a metallic can—and the animal was allowed to interact freely for another 10 min. The memory index was calculated as: [(*novel object interaction time* − *initial object interaction time*)/*total interaction time*]. Values near 1 indicate better memory performance, and values near −1 indicate worse performance.

#### 2.8.4. Reciprocal Interaction Test

After 24 h of social isolation, a rat was randomly paired with a socially novel partner from the same dietary group. Animals freely interacted in a 1 m^2^ sawdust-filled arena for 10 min while being recorded. Two trained, blinded observers scored the recordings for social interaction (total time spent nose-to-nose, nose-anogenital, and side-sniffing) and social disinterest (total time spent self-grooming, digging, lying flat, or freezing near or avoiding the partner). Scored times (s) were averaged. The inter-observer variability was 3.1%.

### 2.9. Sample Collection and Sacrifice

In anesthetized animals (ketamine (100 mg/kg, Richter Pharma AG, Welse, Austria) and xylazine (10 mg/kg, Ecuphar N.V., Oostkamp, Belgium, i.p.), nose-to-anus length and waist circumference were measured. Waist-to-length ratio was calculated. Blood from the abdominal aorta was collected into EDTA and heparin tubes (Sarstedt, Nümbrecht, Germany). The blood count was determined using a hemocytometer (Abacus Vet5, Diatron, Budapest, Hungary). Aliquots of plasma obtained by centrifugation (1600 g/10 min/4 °C, Centrifuge 5804 R, Eppendorf, Hamburg, Germany) were stored at −20 °C. The liver was removed, weighed, and aliquots for analyses of cholesterol [18] and triacylglycerols [19] content were snap-frozen in liquid nitrogen and stored at −80 °C. The olfactory bulbs (OB) and brain were removed, the hypothalamus was dissected, and the tissues snap-frozen in liquid nitrogen were stored at −80 °C. Ten percent homogenates in PBS were prepared using bead-based homogenization (2 cycles at 20 Hz for 1 min).

### 2.10. Plasma Analyses and Calculations

Triacylglycerols (P-TAG), total cholesterol (P-CHOL), high-density (HDL-C) and low-density lipoprotein cholesterol (LDL-C), creatinine and uric acid concentrations, aspartate aminotransferase (AST), and alanine aminotransferase (ALT) activities were measured on a Biolis 24i Premium analyzer (Tokyo Boeki Machinery, Tokyo, Japan). The continuous metabolic syndrome z-score (cMSS) was calculated as: *Z-score (SBP)* − *Z-score (QUICKI)* + *Z-score (waist-to-length ratio)* + *Z-score (P-TAG)* − *Z-score (P-HDL-C)* [20]. The hepatic steatosis index was calculated using a previously published formula [21]. Concentrations of adiponectin, leptin, glucagon-like peptide-1 (GLP-1), corticosterone (all: Crystal Chem; Elk Grove Village, IL, USA), receptor for advanced glycation end products (RAGE), myeloperoxidase (MPO) (both: DuoSet Elisa Kit, R&D Systems, Inc., Minneapolis, MN, USA), N^ε^-(carboxymethyl)lysine (CML), soluble vascular adhesion protein-1 (sVAP-1), pentosidine (all: MyBioSource, Inc.; San Diego, CA, USA), and testosterone (DRG Diagnostic, Marburg, Germany) were quantified using ELISA methods according to manufacturers’ instructions. Advanced oxidation protein products (AOPPs) were measured spectrophotometrically [22]. To evaluate neuroinflammation, levels of glial fibrillary acidic protein (GFAP; GFAP ELISA kit, ELK Biotechnology Co., Ltd., Denver, CO, USA) were quantified in plasma and homogenates of the hypothalamus and olfactory bulb, and tumor necrosis factor-α (TNF-α; DuoSet ELISA Kit, R&D Systems, Inc., Minneapolis, MN, USA) in the homogenates. Plasma levels of inflammatory cytokines (e.g., IL-1α, IL-1β, TNF-α, IL-6, IL-12p70, IL-17A, IL-18, IL-23, GM-CSF, INF-γ, CXCL1/KC, CCL2/MCP-1) were analyzed using the LEGENDplex™ Rat Inflammation Panel (Biolegend, San Diego, CA, USA) in V-bottom plates on a DxFlex cytometer (Beckman, Indianapolis, IN, USA) following the manufacturer’s instructions.

To quantify extracellular DNA (ecDNA) concentration, 100 µL of heparinized plasma obtained at 1600 g (10 min/4 °C) was centrifuged at 16,000× *g* for 10 min at 4 °C (Centrifuge 5804 R, Eppendorf, Hamburg, Germany). EcDNA was isolated from the supernatant using the QIAcube (Qiagen, Heidelberg, Germany) with the QIAmp DNA Blood Mini Kit. The ecDNA concentration was measured using the Qubit dsDNA HS assay (Invitrogen, Carlsbad, CA, USA). To determine the nuclear and mitochondrial fractions of ecDNA, real-time PCR was used, following the previously published protocol [23]. The single radial enzyme diffusion method was used to determine DNase 1 activity, expressed in KU/mL [24].

### 2.11. Statistics

Data distribution was analyzed using the Shapiro-Wilk test. The analysis of variance (ANOVA) with the Bonferroni post-hoc test was used for between-group comparisons. To evaluate body weight, cumulative caloric intake, and an OGTT, a repeated-measures two-way ANOVA with diet and time as independent factors was performed. A 2-factor ANOVA was used to determine whether the combined effects of dietary group and body weight predict behavioral traits. Data are reported as mean ± SD. Statistical analysis was conducted using JASP, and figures were created in GraphPad Prism v. 8.0.1 (GraphPad Software, Inc., San Diego, CA, USA). The *p*-value of <0.05 was considered significant. Multivariate modeling was performed using SIMCA v. 17 software (Sartorius Stedim Data Analytics AB, Umeå, Sweden). A principal component analysis (PCA) was run to identify clusters and outliers, followed by comparisons between diets using an orthogonal projection to latent structures-discriminant analysis (OPLS-DA) to identify discriminatory variables. Variables with variable of importance for the projection (VIP) values ≥ 1.00 were considered as important contributors. The permutation test (100 permutations) was used to validate the model.

## 3. Results

### 3.1. The CAF Diet Robustly Promotes Obesity

The CTRL diet provided 51% of total calories from carbohydrates, 18% from proteins, and 31% from fats; the FFD provided 41%, 18%, and 41%, respectively; and the CAF diet provided approximately 40% of total calories from carbohydrates, 10% from proteins, and 50% from fats. The mean daily caloric intake was higher in both obesogenic diet groups than in the controls (Figure 1A). Food efficiency was higher in the CAF group and only tended to be higher in the FFD group than in the controls (Figure 1B). Compared to controls, rats on the CAF diet had higher body weights from the first month, and those on the FFD from the second month of the experiment (Figure 1C). Moreover, the FFD group had lower body weight than the CAF group from the 3rd month onward. Throughout the experiment, rats on the standard diet gained 111 ± 40 g; those on the FFD 211 ± 77 g (*p* < 0.05 vs. CTRL), and the CAF diet group 316 ± 87 g (*p* < 0.01 vs. FFD, *p* < 0.001 vs. CTRL), Figure 1D. Proxy measure of central obesity—waist-to-length ratio, increased across the groups (CTRL: 0.90 ± 0.08; FFD: 1.02 ± 0.08, *p* < 0.01 vs. CTRL; CAF: 1.06 ± 0.07, *p* < 0.001 vs. CTRL). However, both obesogenic diets induced similar relative adiposity, as measured by total and abdominal fat mass (Figure 1E,F). Increased adiposity was associated with lower lean mass (Figure 1L). DIO was not associated with hypogonadism (Figure 1G); however, FFD rats exhibited elevated plasma corticosterone concentrations (Figure 1H). The CAF group also displayed higher body temperature than the other two groups (Figure 1I). Renal excretion of 5-HIAA was higher in the CAF animals compared to the other two groups (Figure 1J). Higher indoxyl-sulphate concentrations were associated with both obesogenic diets (Figure 1K). The CAF rats had the highest BMC (Figure 1M), while obese rats exhibited lower BMD than controls, with the lowest values observed in the FFD group (Figure 1N).

### 3.2. The CAF Diet Induces a Pre-Diabetic State

Compared with the other two groups, the CAF diet-consuming rats were glucose intolerant, as evidenced by higher FPG, FPI, and AUC under the glucose and insulin curves during the oGTT (Figure 2A,B), and insulin resistant (Figure 2C). In the CAF rats, the absence of a decline in glycemia after the 1st hour of the oGTT, when hepatic glucose production is suppressed (Figure 2A), suggested muscle insulin resistance, which was confirmed by calculating the M-IR index (Figure 2D). In contrast, FFD consumption promoted hepatic insulin resistance (Figure 2E). Plasma GLP-1 levels did not differ between the groups (Figure 2F). Both groups on obesogenic diets exhibited hyperleptinemia (Figure 2G), accompanied by a compensatory rise in adiponectin levels (Figure 2H), resulting in a higher leptin/adiponectin ratio, which reached significance only in the FFD group compared to controls (Figure 2I). We did not observe significant differences in plasma levels of chemically defined AGEs, such as CML or pentosidine (Figure 2J,K), or in their soluble receptors (sRAGE, Figure 2L). However, high urinary D-lactate excretion in the CAF group indirectly reflects high flux of methylglyoxal (MGO, a reactive dicarbonyl, precursor of several AGEs) through the glyoxalase system (Figure 2M). Similar levels of sVAP-1 among the groups do not support the assumption that increased VAP-1 activity contributed significantly to MGO production in our pre-diabetic rats (Figure 2N).

### 3.3. The CAF Diet Alters Lipid Profile and Induces Liver Steatosis

The CAF diet-consuming rats showed higher plasma total and HDL-C levels than the other two groups (Figure 3A and Figure 3B, respectively). The concentration of LDL-C tended to be higher in the CAF group compared to the controls (Appendix A). Rats on the FFD accumulated cholesterol in the liver (Figure 3C). CAF diet consumption was also associated with the highest triacylglycerolemia (Figure 3D). In contrast, the FFD group showed even lower TAG levels than the controls. The liver TAG content mirrored the pattern of plasma TAG (Figure 3E). Plasma AST and ALT activities were higher in rats consuming obesogenic diets than in controls; however, significance was reached only for ALT in the CAF group (Figure 3F and Figure 3G, respectively). Hepatic steatosis index suggests that CAF diet consumption induced liver steatosis (Figure 3H). Plasma uric acid levels were higher in the CAF diet-consuming rats than in the controls (Figure 3I).

### 3.4. FFD and CAF Diet Consumption Are Associated with a Rise in Blood Pressure

Rats on both obesogenic diets displayed higher SBP than controls (Figure 3J), whereas for DBP, significance was observed only in the CAF diet group (Figure 3K).

### 3.5. CAF Diet Consumption Is Associated with a More Severe Cardiometabolic Risk

The continuous metabolic syndrome z-score (cMSS) increased across the groups, but significance was only reached between the CAF and the CTRL groups (Figure 3L).

### 3.6. The Renal Functions Are Unaffected by Consuming Palatable Obesogenic Diets

Plasma creatinine, creatinine clearance, proteinuria, and urinary NGAL were not affected by either obesogenic diet (Appendix A). However, the CAF rats exhibited higher 24-h diuresis and urine osmolality than the controls, whereas the FFD group did not differ (Appendix A).

### 3.7. Markers of Systemic Inflammation Are Inconsistently Affected Under Palatable Obesogenic Diets

White blood cell and lymphocyte counts were similar between the groups (Figure 4A,B). Still, rats on the FFD and CAF diet showed higher neutrophil counts (Figure 4C). Both groups on obesogenic diets displayed higher ecDNA levels; however, a post-hoc test identified significance only between the FFD rats and controls (Figure 4D). No differences were observed in the nuclear and mitochondrial DNA copy numbers (Appendix A). DNase I activity tended to be lower in groups on obesogenic diets (Figure 4E). Activation of neutrophils and macrophages leads to the release of MPO, which, among others, oxidizes albumin, forming AOPPs. Plasma concentration of MPO tended to be higher in FFD and CAF groups than in the controls (Figure 4F), but only the CAF diet-consuming rats showed higher AOPP levels (Figure 4G). None of the 12 plasma inflammatory markers showed a significant between-group difference (Figure 4H).

### 3.8. Palatable Obesogenic Diets Induce Hypothalamic Inflammation

Compared with the controls, rats on FFD and CAF diets displayed higher amounts of GFAP (Figure 4I) and TNF-α (Figure 4J) in the hypothalamus and higher plasma GFAP concentrations (Figure 4M). Concurrently, the quantities of GFAP (Figure 4L) and TNF-α (Figure 4J) in the olfactory bulbs, as well as the circulating levels of TNF-α (Figure 4H), were similar between the groups.

### 3.9. Diet-Induced Obesity Affects Locomotory Activity, Induces Anxiety-like Behavior and Social Deficits, but Does Not Affect Short-Term Memory

#### 3.9.1. The Open Field Test

The post-hoc test in one-way ANOVA indicated a significant difference between the CTRL and the CAF groups in traveled distance (Figure 5A, *p* < 0.01) and time spent in the central zone (indicating anxiolytic-like behavior, Figure 5B, *p* < 0.05). We conducted a 2-factor ANOVA to evaluate the impact of body weight. The overall models remained significant (*p* < 0.05, both), but they could not distinguish between the effects of diet and body weight, indicating that the impact of “dietary group” is primarily mediated by body weight.

#### 3.9.2. The Elevated-Plus Maze Test

All three groups spent a similar time in the open arms (Figure 5C). In line with travelling a shorter distance (Figure 5D), the CAF rats spent a shorter time moving (Figure 5E) than the controls.

#### 3.9.3. Reciprocal Interaction Test

Both groups on obesogenic diets showed lower social interest (Figure 5F) and higher disinterest (Figure 5G) towards a novel partner from the same dietary group. These were the only behavioral tests for which 2-way ANOVA indicated a significant effect of diet (*p* < 0.001 and *p* < 0.01, respectively), while the independent effect of body weight was insignificant.

#### 3.9.4. The Novel Object Recognition Test

In both testing phases T1 and T2, animals consuming the CAF diet exhibited lower locomotor activity than controls (Figure 5H); however, they spent a similar amount of time sniffing the objects (Figure 5I). We found no differences in the short-term memory index, which varied by approximately 0, indicating that both groups preferred sniffing the new object (Figure 5J).

### 3.10. Between-Group Comparison of the Effects of Obesogenic Diets

We used the OPLS-DA to distinguish the similar and specific effects of two obesogenic diets. Selected markers identified by the ANOVA as significantly differing from the controls in at least one group (FFD and/or CAF) were entered as predictors (independent variables). The biplot of scores and loadings (Figure 6) shows a clear separation of the FFD and CAF groups: individual observations (scores) are visualized as green (FFD, scattered to the left from the origin) and orange rectangles (CAF, scattered to the right). Separation of dietary groups occurs in the horizontal direction; the vertical direction expresses within-group variability (which does not discriminate between the groups). Loadings (markers, in grey) located close to the plot origin (within or near the inner circle) poorly describe between-group differences. Observations near the markers are high in these variables and low in the variables situated opposite. Liver TAG, body temperature, SBP, QUICKI, plasma corticosterone, AOPP/Alb, BMD, urinary D-lactate, plasma TAG, plasma CHOL, and urinary IS significantly contributed to between-group separation (VIP values ranging from 1.46 to 1.04, Table 1). The model indicated that two groups of rats on obesogenic diets did not differ significantly in waist-to-length ratio, plasma GFAP, liver CHOL, urinary excretion of 5-HIAA, plasma adiponectin, social interest, hypothalamic GFAP and TNF-α, plasma uric acid, neutrophil count, and plasma leptin levels. Variables identified by the OPLS-DA model as significantly differing between the groups were higher in the CAF vs. FFD group, except for liver TAG content, QUICKI, plasma corticosterone, and BMD, which were higher in the FFD group than in the CAF diet-consuming animals. The effects on body temperature, QUICKI, corticosterone, AOPP/Alb, D-lactate, plasma and liver CHOL, urinary excretion of 5-HIAA, and uric acid were specific to a single diet; whereas the dietary effects on SBP, bone mineral density, plasma TAG, IS, waist-to-length ratio, and plasma GFAP levels were similar, but the magnitude of the effect differed. Most variables identified by OPLS-DA as not contributing to the separation between the FFD and CAF groups showed similar differences vs. controls. The model accounted for 40% of the variance in R^2^ and 98% of that in R^2^Y, with a predictive accuracy of 92%. The permutation test results (*R*^2^: 0.0; 0.61; *Q*^2^: 0.0; −0.55) showed that the model was not overfit, confirming its accuracy.

## 4. Discussion

This study investigated the long-term metabolic and behavioral effects of two Western-style obesogenic diets consisting of palatable human foods with attractive smell and taste in female rats. The FFD consisted of a fixed combination of items (ground cheeseburgers and fries) that were administered daily, whereas the CAF diet-consuming rats could self-select from rotating menus. Self-selection revealed a preference for fats over proteins. After 12 months, FFD induced moderate obesity, whereas CAF produced severe obesity, though total and abdominal fat proportions were comparable. Both diets caused neuroinflammation and social deficits, likely mediated by dysbiotic microbiota that altered tryptophan metabolism, but elicited distinct organ-specific metabolic pathologies. Key overlaps and differences identified by OPLS-DA are summarized in Table 1 and discussed below.

In rodents, DIO progressively develops into a metabolic syndrome–like phenotype, characterized by increased (central) adiposity, hyperglycemia, hyperinsulinemia, systemic insulin resistance, dyslipidemia (elevated fasting TAG and reduced HDL-C), elevated BP, and excessive intrahepatic TAG accumulation.

### 4.1. Different Cardio-Metabolic Syndrome-like Traits in DIO Models

CAF animals gained more weight than FFD ones due to higher caloric intake and food efficiency, although total and abdominal fat percentages, as well as waist-to-length ratios (a proxy for central obesity), were similar between the two groups. That the CAF diet is more robust than other obesogenic diets in inducing obesity is consistent across studies [11,25,26,27]. The CAF diet-consuming rats massively gained weight, albeit they displayed an elevated rectal temperature, implying increased thermogenesis. There is no consensus on whether brown adipose tissue [28], or the liver [29] is the primary driver of CAF diet-induced heat production. Concurrently, CAF diet-fed rats exhibited higher urinary 5-HIAA excretion, indicating enhanced peripheral serotonergic activity, which can serve as an anabolic signal to conserve energy, among others, by suppressing brown fat thermogenesis [30]. We have only a single rectal temperature measurement, so it is unclear whether brown adipose thermogenesis drives the higher body temperature. Even so, the anabolic action of peripheral serotonin likely cannot offset the thermogenesis triggered by the CAF diet, which, in turn, does not prevent diet-induced weight gain. Longitudinal tracking of brown fat activity (including uncoupling protein-1 expression), body temperature, and peripheral serotonin metabolism initiated early in the development of obesity could clarify how these systems interact over time.

Rats fed the FFD developed moderate obesity with elevated blood pressure, whereas CAF-fed rats exhibited severe obesity, hypertension, and insulin resistance (QUICKI). Both diets increased circulating leptin and adiponectin levels, while testosterone levels remained unchanged. Hyperleptinemia reflected excess adiposity, likely due to leptin resistance [31]. In contrast to the typically reduced adiponectin levels observed in obesity [32], both diet groups exhibited elevated concentrations, consistent with recent findings in CAF-fed male mice [33], although the mechanisms underlying this compensatory response remain unclear.

Associations between DIO and BP are inconsistent and appear sex-dependent. In male rats, a lard-based HFD induced hypertension, visceral obesity, and insulin resistance, effects absent with an isocaloric vegetable-oil-containing HFD [34], indicating that fatty acid composition rather than caloric load drives these outcomes. Gonadectomy prevented hypertension, which testosterone restored. Female rats, intact or ovariectomized, did not develop hypertension despite insulin resistance. In male mice, the HF or CAF diets increased visceral adiposity, glucose, insulin, and leptin without affecting BP or weight gain [12], while CAF-fed female rats showed increased adiposity and leptin without BP elevation [35]. Coatmellec-Taglioni et al. [36] further reported sex dimorphism: both sexes developed adiposity and hyperleptinemia, but only males manifested hypertension, possibly due to alterations in renal adrenergic and leptin receptors. In our study, FFD-fed females developed elevated BP, and CAF-fed females became hypertensive. However, single-time-point measurements of BP, leptin, and insulin sensitivity limit interpretation of their interactions. Notably, kidney function—assessed by creatinine clearance, protein, and NGAL excretion—was similar across groups, suggesting that obesity-related kidney disease is unlikely to account for the elevated BP. Future research should employ longitudinal designs to investigate sex differences and dynamic relationships among BP, insulin sensitivity, and leptin levels in DIO models.

After 12 months, CAF-fed rats exhibited metabolic syndrome–like phenotype, including hyperglycemia, hyperinsulinemia, glucose intolerance, systemic (QUICKI) and muscle insulin resistance, general and central obesity, hypertension, elevated plasma and liver TAG, and elevated plasma cholesterol. In contrast, FFD-fed rats developed a distinct profile characterized by elevated BP and adiposity, but without glucose intolerance and systemic insulin resistance. Instead, they showed an inverted steatotic pattern, characterized by low plasma and hepatic TAG yet high hepatic cholesterol accumulation, suggesting differential regulation of TAG and cholesterol homeostasis under FFD-induced stress, reflected by hypercorticosteronism. The insulin-sensitive obese phenotype may reflect adipose hyperplasia and smaller, more insulin-sensitive adipocytes rather than adipocyte hypertrophy [37], while hepatic cholesterol overload likely results from excessive dietary intake combined with impaired clearance mechanisms [38]. Future work should validate this phenotype through hepatic gene expression, lipoprotein flux, and lipidomic analyses to delineate regulatory differences in fatty acid oxidation and cholesterol handling.

Unlike in humans, there are no standardized cut-offs for diagnosing metabolic syndrome in rodents. This limitation can be addressed by calculating a continuous metabolic syndrome z-score, which, in our study, revealed markedly higher cardiometabolic risk in CAF-fed rats compared with FFD-fed rats. However, as z-scores are population-specific, cross-study comparisons are not valid.

### 4.2. Carbonyl Stress Markers in DIO Models

To assess cumulative protein damage from the Maillard reaction, we quantified two AGEs, CML and pentosidine. These arise from the modification of lysine or arginine by reactive sugars, α-dicarbonyls, or glyoxidation [39,40]. AGE-modification alters protein structure and function, and the interaction with RAGE promotes inflammation, oxidative stress, and atherogenesis, contributing to the pathogenesis of non-communicable diseases, including obesity-associated metabolic disorders [41,42].

Plasma levels of endogenous (protein-bound) AGEs—CML and pentosidine—did not differ between the groups, suggesting no overt increase in glycoxidative or carbonyl stress. However, prior studies have reported elevated CML in HFD-fed male rats [10], whereas human obesity is paradoxically associated with low plasma CML, likely due to adipose sequestration, in which CML dysregulates adipokines and induces insulin resistance [43]. In healthy, medication-free adults, CML levels inversely correlate with all components of metabolic syndrome, except HDL-C (which shows a direct relationship) [44]. Neither pentosidine (reflecting collagen turnover and remodeling) is elevated in the plasma of subjects with obesity [45]. However, HFD-fed mice, particularly females, accumulate CML and pentosidine in their bones [46]. Thus, tissue rather than plasma AGE levels may better capture glycoxidative/carbonyl stress in DIO.

CAF-fed rats exhibited elevated urinary D-lactate, consistent with reports on higher plasma levels in obese humans and mice [47]. In mammals, D-lactate is formed by the glyoxalase pathway from a glycolytic by-product, methylglyoxal [40]. In humans, D-lactate is more closely associated with obesity than with insulin resistance [48], which may explain its rise during CAF feeding. We also measured sVAP-1, since its semicarbazide-sensitive amine oxidase (SSAO) activity generates methylglyoxal, and hyperglycemia-induced VAP-1 produces H_2_O_2_ with insulin-sensitizing effects [49]. However, our data did not implicate VAP-1/SSAO in methylglyoxal production or insulin sensitivity. Alternatively, elevated D-lactate excretion may reflect gut colonization by D–lactate–producing microbes [50], underscoring the need for future studies targeting the microbiota during CAF diet interventions.

### 4.3. Metainflammation in DIO Models

Diet-induced obesity is characterized by chronic, low-grade systemic and neuroinflammation [51], which persists without increased energy expenditure and thereby coexists with weight gain. Visceral adipose tissue functions as an endocrine organ, releasing adipokines, cytokines, and other bioactive mediators that promote sterile inflammation, insulin resistance, endothelial dysfunction, and cardiovascular pathology, while locally recruiting pro-inflammatory immune cells, which further amplify cytokine production [52,53]. In our study, obesogenic diets did not elevate plasma cytokine or chemokine levels. Evidence from rodent models suggests a time-dependent systemic inflammatory response: In male rats, short-term CAF diet feeding increased plasma TNF-α [53], whereas longer CAF or HFD exposures did not [10,54].

Both obesogenic diet groups showed inflammatory markers, such as higher neutrophil counts, while CAF rats additionally exhibited higher AOPP levels, indicating phagocyte activation, since AOPPs are formed on albumin via myeloperoxidase reaction [55]. Although MPO levels were similar across groups, their short half-lives limit interpretation; a persistent elevation of AOPPs would reflect chronic phagocyte activation. Obese rats also displayed increased ecDNA, which is released during cell death or NET formation. In obesity, the majority of ecDNA originates from hematopoiesis, adipose tissue, the spleen, and the pancreas [56]. Undegraded autologous ecDNA can act as a danger-associated molecular pattern (DAMP), driving sustained inflammation that may perpetuate a self-amplifying cycle [57]. While some studies in healthy adults link ecDNA to BMI or visceral fat area [58], in healthy adolescents, ecDNA was related to cardiometabolic risk rather than obesity per se [59]. Both diet groups also developed hyperleptinemia, which exacerbates adipogenesis and immune cell activation, further amplifying metabolic and inflammatory disturbances.

### 4.4. Hypothalamic Neuroinflammation in DIO Models

In the central nervous system (CNS), IS activates the aryl hydrocarbon receptor, disrupting the blood–brain barrier and impairing cognition [60]. IS, or sera from patients with chronic kidney disease, induce oxidative stress and neuroinflammation in glial cells and astrocytes, effects attenuated by the IS adsorbent [61].

The olfactory nerve and OB are CNS structures that process odor signals. In the OB, astrocytes convert the chemical signals into neural signals for odor perception. Unlike the hypothalamus, TNF-α and GFAP levels remained similar in OBs of our rats, likely reflecting the need to maintain stable olfactory processing for survival, even under conditions of metabolic stress.

In rodent DIO models, neuroinflammation precedes body weight gain and peripheral inflammation, with the hypothalamus affected early [51]. An increase in hypothalamic GFAP and TNF-α in our study suggests that an obesogenic diet induces a shift in astrocytes towards astrogliosis, characterized by the production of inflammatory cytokines, regardless of the severity of the metabolic syndrome-like phenotype. Concomitant elevation of plasma GFAP suggests its potential as a biomarker of astrocyte activation.

Although hypothalamic GFAP and TNF-α are established indicators of neuroinflammation in metabolic models, they do not capture the complexity of glial activation, including microglial dynamics, cytokine signaling, and transcriptional reprogramming. Future studies should investigate additional molecular and histological markers and employ transcriptomic profiling to obtain an integrated picture of neuroinflammatory processes.

### 4.5. Tryptophan Metabolites in DIO Models

Hyperphagia and gut microbiota dysbiosis are key drivers of DIO. The CAF diet is a robust model that induces adiposity through hedonic feeding and alterations in gut microbiota [10,62,63,64]. The fixed-item FFD has a lower incentive value, resulting in lower energy intake and lower weight gain. Unlike the CAF diet, the impact of FFD on microbiota is unknown, though our data on 5-HIAA and IS suggest distinct effects.

Obesity-associated gut dysbiosis increases energy harvest, enhances lipid synthesis and storage, increases gut permeability, production of metabolites that affect satiety, promotes insulin resistance, and sustains low-grade inflammation [65]. Microbiota transplantation studies confirm its causal role [66,67]. Disruption of gut–brain satiety signaling further promotes hyperphagia and weight gain. Diet appears to be the primary determinant, as HF and CAF feeding rapidly reduce microbial diversity before obesity and insulin resistance develop [10,64,68]. Although we did not assess microbiota composition, elevated urinary IS and 5-HIAA in obesogenic diet-fed rats indicate altered tryptophan metabolism consistent with microbial involvement.

Colonic bacteria convert dietary tryptophan to indole, which is metabolized in the liver to IS, further excreted by the kidneys. Gut microbiota composition determines IS production following an oral tryptophan load [69]. IS is a uremic toxin linked to multiorgan toxicity [70,71], seldom studied in non-renal pathologies. Our FF- and CAF diet-fed rats excreted more IS than the controls, consistent with prior reports in HFD-fed rodents [72], reflecting that HF and high-carbohydrate diets shift microbiota towards indole overproduction [73]. Both the HFD and IS disrupt gut barrier integrity, enhancing systemic absorption of indole [74]. Thus, IS promotes its own toxicity by enhancing the absorption of its precursor [75]. Additionally, the rate-limiting enzyme in IS formation, CYP2E1, is upregulated in liver steatosis and circulating IS levels increase with rising triacylglycerolemia. The FFD rats exhibited low TAG levels; thus, high urinary IS likely reflected increased microbial indole supply rather than hepatic conversion. Experimental [72] and clinical data [76,77] support a link between IS and glucose metabolism.

In the periphery, mainly enterochromaffin cells (EC) produce serotonin from tryptophan, in response to luminal nutrients [78]. Spore-forming bacteria (e.g., Clostridia and Bacilli, which increase under CAF feeding) also promote serotonin production [79]. To regulate serotonergic signaling, enterocytes and platelets sequester serotonin. Activated thrombocytes release serotonin, which induces local vasoconstriction, smooth muscle proliferation, and aggregation. Therefore, free serotonin is rapidly metabolized to 5-HIAA and excreted via the kidneys, serving as a marker of peripheral serotonin production [80]. In the gut, serotonin modulates motility, enhances nutrient absorption, and promotes energy storage by stimulating insulin release, promoting lipogenesis, and suppressing lipolysis and thermogenesis [30]. The higher urinary excretion of 5-HIAA in CAF animals more likely reflects microbiota differences than renal retention or dietary intake, as the CAF diet lacked serotonin-rich foods. In humans, obesity and metabolic syndrome are associated with elevated serotonin release and circulating or urinary 5-HIAA levels, which correlate with BMI, glycemia, HbA1c, BP, lipid profile, uric acid, and inflammatory markers [81,82]. Our findings point to similar associations. Experimental evidence indicates that 5-HIAA is a metabolically active end-product impairing muscle glucose uptake [83]. Together, these findings suggest that nutrient- and microbiota-induced increases in peripheral serotonin synthesis may contribute to obesity-related metabolic complications, including liver steatosis.

Elevated urinary IS unequivocally indicates, and higher 5-HIAA excretion strongly suggests, alterations in microbial tryptophan metabolism. As we did not directly assess gut microbiota composition, the involvement of dysbiosis in these changes remains inferential. Future studies combining microbiome sequencing with targeted metabolomics are warranted to confirm this link.

### 4.6. Behavioral Effects of DIO Models

Reduced locomotor activity and anxiety-like behavior probably reflect adiposity-related motor constraints. While some studies report an anxiolytic-like phenotype in DIO [26,84], early CAF exposure induces anxiety-like behavior in juveniles [26]. Differences in microbiota-derived metabolites (5-HIAA, IS) and diet composition between CAF and FFD suggest distinct contributions of the gut–brain axis to behavior [85]. We observed no short-term memory deficits, although spatial learning impairments have been reported in CAF and HFD models [86,87]. Social deficits occurred in both DIO groups, independent of adiposity, implicating amygdala- and prefrontal cortex-dependent pathways [86,88]. CAF withdrawal has been linked to monoaminergic and endocannabinoid changes in reward circuits, provoking emotional disturbances [89]. Given the association of social deficits with evoked and emotional eating and the possibility that food-seeking may override sociability in reciprocal interaction tests, future studies should investigate whether these deficits in DIO arise from endocannabinoid signaling, evoked eating, or shared neuroinflammatory mechanisms.

### 4.7. Limitations

Due to technical and budgetary constraints associated with the 12-month duration of the experiment, we studied only intact female rats, aiming to model physiological hormonal conditions. This approach helped counter the male bias common in preclinical obesity research. Future studies that include both sexes are warranted to gain a comprehensive understanding of DIO-specific pathologies and to improve translational relevance. Another limitation is the cross-sectional design, which precludes the determination of cause-and-effect relationships between obesity development, neuroinflammatory, and behavioral responses. Repeated anesthesia, interim placement in metabolic cages, and behavioral testing requiring 2–3 days are known to induce short-term weight loss and transient changes in inflammatory proteins, while repeated behavioral testing can also lead to habituation and learning effects. Therefore, these interventions were intentionally avoided to prevent bias in the longitudinal course. Future studies using serial biomarker assessments are warranted to define temporal dynamics. Our study lacks histopathological tissue assessment, which would enable a more objective evaluation of the outcomes of different Western-type diet consumption. It also lacks microbiome profiling, which precludes direct attribution of metabolite alterations to specific microbial taxa or pathways. Future research should integrate microbiome and metabolomic analyses to delineate causal links between gut dysbiosis, tryptophan metabolism, D-lactate production, and neuroinflammatory outcomes. Neuroinflammatory changes should be objectivized using broader panels, including microglial markers, cytokine arrays, or in situ imaging. Furthermore, energy metabolism should be investigated using objective methods, such as indirect calorimetry and the assessment of uncoupling protein expression. Additives influence the outcomes of Western-type diets in the foods consumed. In our view, their presence in both Western-type diets objectively reflects real-world dietary human exposure.

## 5. Conclusions

Our findings highlight that, except for caloric load and diet composition, feeding behavior determines the trajectory of obesity-associated pathologies. Future studies incorporating longitudinal cardiometabolic assessments, microbiome analyses, and tissue-level pathology will be critical for disentangling diet-specific mechanisms and refining translational models of human obesogenic diets.

## Figures and Tables

**Figure 1 nutrients-17-03614-f001:**
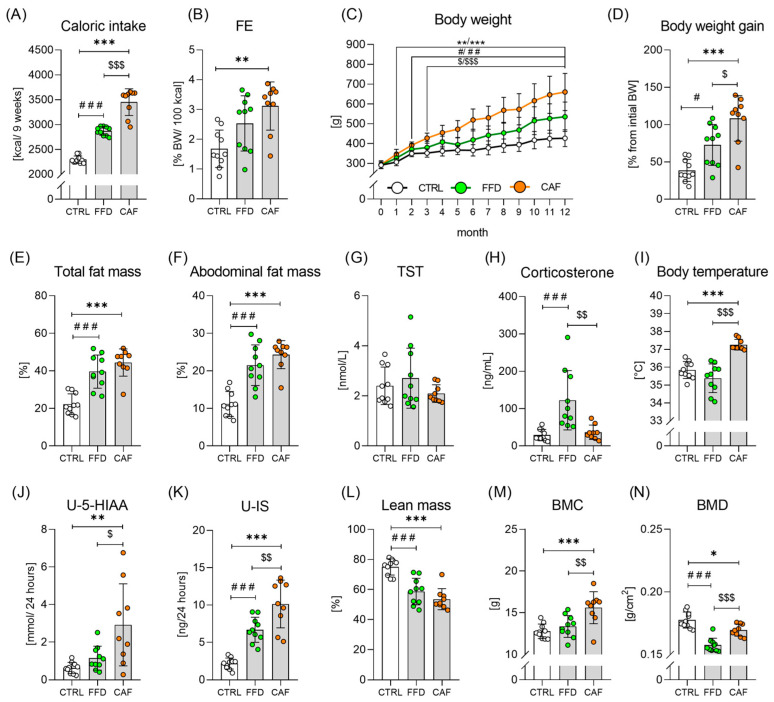
**Obesity development in fast-food and cafeteria diet groups.** (**A**) Total caloric intake during 9 monitored weeks (F = 117.9; *p* < 0.001); (**B**) Food efficiency (F = 4.6; *p* < 0.05); (**C**) Body weight progress over 12 months (F_time_= 49.7; *p* < 0.001; F_diet_ = 24.9; *p* < 0.001; F_interaction_ = 7.8; *p* < 0.001); (**D**) Body weight gain (F = 18.5; *p* < 0.001); (**E**) Total fat mass (F = 24.6; *p* < 0.001); (**F**) Abdominal fat mass (F = 25.4; *p* < 0.001); (**G**) Plasma testosterone concentration (F = 1.2; *p* = *ns*); (**H**) Plasma corticosterone concentration (F = 11.5; *p* < 0.001); (**I**) Body temperature (F = 27.2; *p* < 0.001); (**J**) Urinary concentration of 5-hydroxy-indol acetic acid (5-HIAA; F = 8.3; *p* < 0.01); (**K**) Urinary concentration of indoxyl sulphate (F = 35.3; *p* < 0.001); (**L**) Lean mass (F = 4.6; *p* < 0.05); (**M**) Bone mineral content (BMC; F = 10.6; *p* < 0.001); (**N**) Bone mineral density (BMD; F = 18.8; *p* < 0.001). Control chow group (CTRL; n = 10); fast-food diet group (FFD; n = 10); cafeteria diet group (CAF; n = 9). *—CAF vs. CTRL; #—FFD vs. CTRL; $—CAF vs. FFD */#/$—*p* < 0.05; **/##/$$—*p* < 0.01; ***/###/$$$—*p* < 0.001. Data are presented as mean ± SD.

**Figure 2 nutrients-17-03614-f002:**
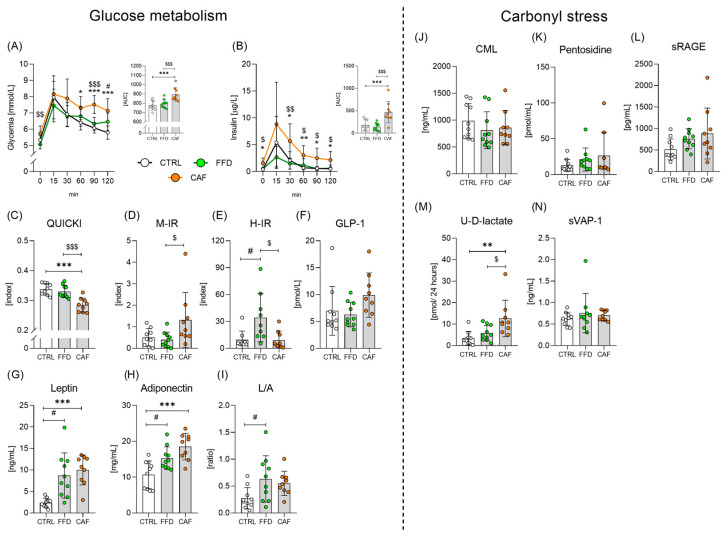
**Glucose metabolism and carbonyl stress.** (**A**) Glycemic curves from Oral glucose tolerance test (F _time_= 60. 4; *p* < 0.001; F_diet_= 10.3; *p* < 0.001; F_interaction_ = 2.5; *p* < 0.01) and area under the glycemic curve (AUC; F = 12.4; *p* < 0.001); (**B**) Insulin curves from Oral glucose tolerance test (F _time_= 19.1; *p* < 0.001; F_diet_= 10.1; *p* < 0.001; F_interaction_ = 2; *p* < 0.05) and area under the insulin curve (AUC; F = 12.1; *p* < 0.001); (**C**) The quantitative insulin sensitivity check index (QUICKI; 15.4; *p* < 0.001); (**D**) Muscle insulin resistance index (M- IR; F = 3.8; *p* < 0.05); (**E**) Hepatic insulin resistance index (H-IR; F = 6.1; *p* < 0.01); (**F**) Glucagon-like peptide 1 (GLP-1; F = 2.4; *p* = *ns*); (**G**) Plasma leptin concentration (F = 11.9; *p* < 0.001); (**H**) Plasma adiponectin concentration (F = 11.5; *p* < 0.001); (**I**) Leptin- adiponectin ratio (L/A; F = 3.7; *p* < 0.05); (**J**) Plasma carboxymethyl lysin concentration (CML; F = 0.7; *p* = *ns*); (**K**) Plasma pentosidine concentration (F = 1.1; *p* = *ns*); (**L**) Concentration of soluble receptor for advanced glycation end products (sRAGE; F = 2.3; *p* = *ns*); (**M**) Urinary concentration of D-lactate (U-D-lactate; F = 4.3; *p* < 0.05); (**N**) Plasma concentration of soluble vascular adhesion protein-1 (sVAP-1; F = 0.5; *p* = *ns*). Control chow group (CTRL; n = 10); fast-food diet group (FFD; n = 10); cafeteria diet group (CAF; n = 9). *—CAF vs. CTRL; #—FFD vs. CTRL; $—CAF vs. FFD */#/$—*p* < 0.05; **/$$—*p* < 0.01; ***/$$$—*p* < 0.001. Data are presented as mean ± SD.

**Figure 3 nutrients-17-03614-f003:**
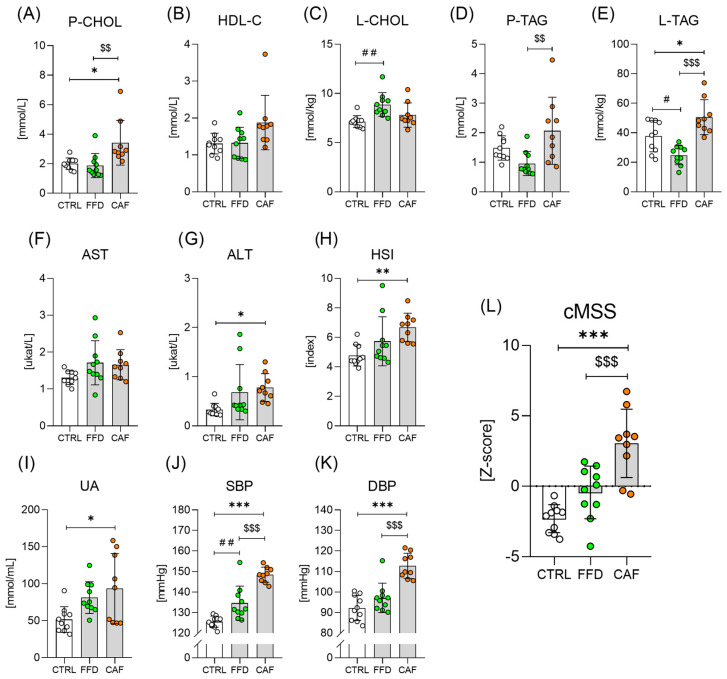
**Lipid profile, markers of liver steatosis, and cardiometabolic risk markers.** (**A**) Plasma concentration of cholesterol (P-CHOL; F = 6.9; *p* < 0.01); (**B**) Plasma concentration of high-density lipoprotein cholesterol (HDL-C; F = 3.8; *p* < 0.05); (**C**) Liver cholesterol content (L-CHOL; F = 16.2; *p* < 0.001); (**D**) Plasma concentration of triacylglycerols (P-TAG; F = 5.7; *p* < 0.01); (**E**) Liver triacylglycerol content (L-TAG; F = 6.8; *p* < 0.01); (**F**) Plasma aspartate aminotransferase activity (AST; F = 2.6; *p* = *ns*); (**G**) Plasma alanine aminotransferase activity (ALT; F = 3.9; *p* < 0.05); (**H**) Hepatic steatosis index (HIS; F = 6.1; *p* < 0.01); (**I**) Plasma uric acid concentration (F = 4.7; *p* < 0.05); (**J**) Systolic blood pressure (SBP; F = 40.8; *p* < 0.001); (**K**) Diastolic blood pressure (F = 25.3; *p* < 0.001); (**L**) Continuous metabolic syndrome Z-score (cMSS; F = 20.6; *p* < 0.001). Control chow group (CTRL; n = 10); fast-food diet group (FFD; n = 10); cafeteria diet group (CAF; n = 9). *—CAF vs. CTRL; #—FFD vs. CTRL; $—CAF vs. FFD */#—*p* < 0.05; **/##/$$—*p* < 0.01; ***/$$$—*p* < 0.001. Data are presented as mean ± SD.

**Figure 4 nutrients-17-03614-f004:**
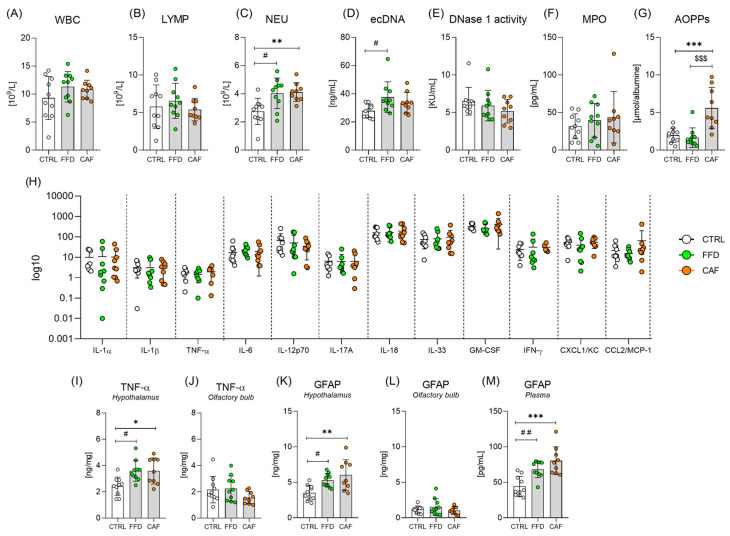
**Systemic inflammation and neuroinflammation**. (**A**) White blood cells (WBC; F = 1.3; *p* = *ns*); (**B**) Lymphocytes (LYMP; F = 0.6; *p* = *ns*); (**C**) Neutrophils (NEU; F = 6.9; *p* < 0.01); (**D**) Plasma extracellular DNA concentration (ecDNA; F = 3.7; *p* < 0.05); (**E**) DNase 1 activity (F = 1.2; *p* = *ns*); (**F**) Plasma myeloperoxidase concentration (MPO; F = 0.6; *p* = *ns*); (**G**) Plasma advanced oxidation protein products concentration (AOPPs; F = 13.8; *p* < 0.001); (**H**) Inflammatory panel concentration: Interleukin (IL)-1 α (F = 0.008; *p* = *ns*); IL-1β (F = 0.008; *p* = *ns*); Tumor necrosis factor α (TNF-α; F = 0.4; *p* = *ns*); IL-6 (F = 0.3; *p* = *ns*); IL-12p70 (F = 0.6; *p* = *ns*); IL-17A (F = 0.002; *p* = *ns*); IL-18 (F = 0.2; *p* = *ns*); IL-33 (F = 0.2; *p* = *ns*); Granulocyte-Macrophage Colony-Stimulating Factor (GM-CSF; F = 0.9; *p* = *ns*); Interferon γ (IFNγ; F = 0.1; *p* = *ns*); C-X-C motif chemokine ligand 1 (CXCL1; F = 0.6; *p* = *ns*); Chemokine (C-C motif) ligand 2/Monocyte Chemoattractant Protein-1 (CCL2/MCP-1; F = 1.1; *p* = *ns*); (**I**) Hypothalamic TNF-α concentration (F = 6.4; *p* < 0.01) (**J**) TNF-α concentration in the olfactory bulb (F = 1.8; *p* = *ns*); (**K**) Hypothalamic glial fibrillary acidic protein (GFAP) concentration (F = 7.5; *p* < 0.01); (**L**) GFAP concentration in olfactory bulb (F = 0.9; *p* = *ns*); (**M**) Plasma concentration of GFAP (F = 13.7; *p* < 0.001). Control chow group (CTRL; n = 10); fast-food diet group (FFD; n = 10); cafeteria diet group (CAF; n = 9). *—CAF vs. CTRL; #—FFD vs. CTRL; $—CAF vs. FFD */#—*p* < 0.05; **/##—*p* < 0.01; ***/$$$—*p* < 0.001. Data are presented as mean ± SD.

**Figure 5 nutrients-17-03614-f005:**
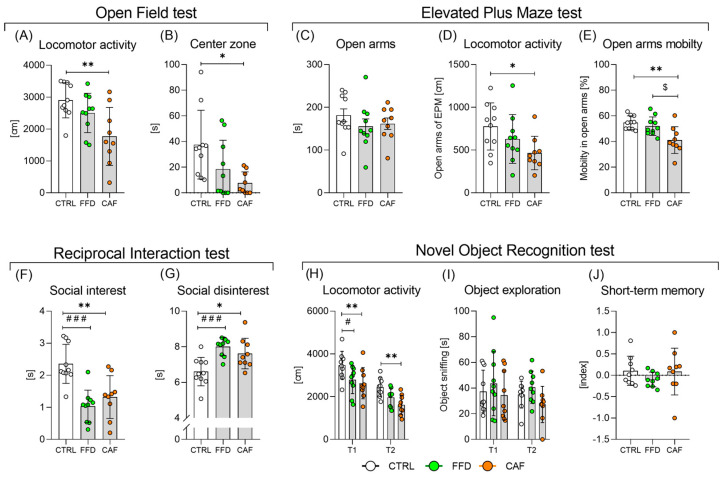
**Behavioral consequences of obesogenic diets**. (**A**) Locomotor activity (F = 6.3; *p* < 0.01); (**B**) Time spent in the central zone of Open Field test (F = 4.9; *p* < 0.05); (**C**) Time spent in the open arms of Elevated Plus maze test (F = 0.8; *p* = *ns*); (**D**) Locomotor activity in the open arms of Elevated Plus maze test (F = 3.5; *p* < 0.05); (**E**) Percentage of mobility in the open arms of Elevated Plus maze test (F = 7.9; *p* < 0.01); (**F**) Social interest (F = 13.6; *p* < 0.001); (**G**) Social disinterest (F = 9.8; *p* < 0.001); (**H**) Locomotor activity in the Novel Object recognition test (F_group_= 6.2; *p* < 0.01; F_trial_= 149.8; *p* < 0.001; F _interaction_= 1.1; *p* = *ns*); (**I**) Exploration behavior in the Novel Object Recognition test (F_group_= 1.5; *p* = *ns*; F_trial_= 1.4; *p* = *ns*; F _interaction_= 0.2; *p* = *ns*); (**J**) Short-term memory index (F = 0.9; *p* = *ns*). Control chow group (CTRL; n = 10); fast-food diet group (FFD; n = 10); cafeteria diet group (CAF; n = 9). *—CAF vs. CTRL; #—FFD vs. CTRL; $—CAF vs. FFD */#/$—*p* < 0.05; **—*p* < 0.01; ###—*p* < 0.001. Data are presented as mean ± SD.

**Figure 6 nutrients-17-03614-f006:**
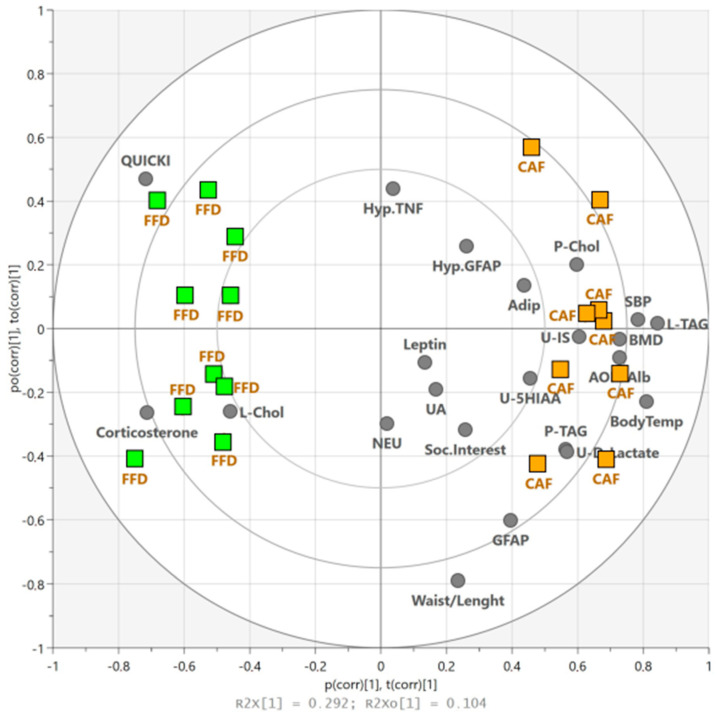
**The biplot co-charting scores and loadings.** The scores represent new variables that summarize information from all independent variables, so that one score vector corresponds to one animal, having its own unique score vector. The model shows complete separation (in the *x*-axis direction) between the FFD (green squares) and CAF (orange squares) groups. The separation along the *y*-axis represents within-group variability. Grey circles represent loadings (independent variables). Those situated in the vicinity of green squares are high in the FFD group, while those situated near the orange squares are higher in the CAF than FFD rats. Independent variables within and near the central ellipse do not contribute to between-group separation. FFD, fast-food diet; CAF, cafeteria diet; 5HIAA, urinary 5-hydroxy-3-indoleacetic acid; Adip, adiponectin; AOPP/Alb, advanced oxidation protein products corrected for plasma albumin; Body Temp, Body temperature; BMD, bone mineral density; GFAP, glial fibrillary acidic protein; Hyp., hypothalamic; IS, indoxyl sulphate; L-, liver; NEU, neutrophils; QUICKI, quantitative insulin sensitivity check index; SBP systolic blood pressure; Soc., social; TAG, triacylglycerols; TNF, tumor necrosis factor-α; U-, urinary excretion.

**Table 1 nutrients-17-03614-t001:** Comparison of the effects of fast-food and cafeteria diets on variables included in the OPLS-DA model.

FFD vs. CAF			Obesogenic diets vs. CTRL
	VIP	Difference	
Triacylglycerols (liver)	** *1.46* **	**↓ CAF**	CAF: ↑ vs. CTRL; FFD: ↓ vs. CTRL
Body temperature	** *1.42* **	**↑ CAF**	CAF: ↑ vs. CTRL
Systolic blood pressure	** *1.35* **	**↑ CAF**	CAF, FFD: ↑ vs. CTRL
QUICKI	** *1.34* **	**↓ CAF**	CAF: ↓ vs. CTRL
Corticosterone (plasma)	** *1.27* **	**↓ CAF**	FFD: ↑ vs. CTRL
AOPPs (plasma)	** *1.26* **	**↑ CAF**	CAF: ↑ vs. CTRL
Bone mineral density	** *1.26* **	**↓ CAF**	CAF, FFD: ↓ vs. CTRL
D-lactate (urine)	** *1.07* **	**↑ CAF**	CAF: ↑ vs. CTRL
Triacylglycerols (plasma)	** *1.06* **	**↑ CAF**	CAF, FFD: NS vs. CTRL
Cholesterol (plasma)	** *1.05* **	**↑ CAF**	CAF: ↑ vs. CTRL
Indoxyl Sulphate (urine)	** *1.04* **	**↑ CAF**	CAF, FFD: ↑ vs. CTRL
Waist-length ratio	*0.98*	NS	CAF, FFD: ↑ vs. CTRL
GFAP (plasma)	*0.96*	NS	CAF, FFD: ↑ vs. CTRL
Cholesterol (liver)	*0.85*	NS	FFD: ↑ vs. CTRL
5-HIAA (urine)	*0.80*	NS	CAF: ↑ vs. CTRL
Adiponectin (plasma)	*0.77*	NS	CAF, FFD: ↑ vs. CTRL
Social interest	*0.57*	NS	CAF, FFD: ↓ vs. CTRL
GFAP (hypothalamus)	*0.54*	NS	CAF, FFD: ↑ vs. CTRL
TNF-α (hypothalamus)	*0.50*	NS	CAF, FFD: ↑ vs. CTRL
Uric acid (plasma)	*0.36*	NS	CAF: ↑ vs. CTRL
Neutrophil count	*0.34*	NS	CAF, FFD: ↑ vs. CTRL
Leptin (plasma)	*0.26*	NS	CAF, FFD: ↑ vs. CTRL

↑—higher, ↓—lower.

## Data Availability

All data obtained are given within the article.

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
