# Peer review of "Cafeteria and Fast-Food Diets Induce Neuroinflammation, Social Deficits, but a Different Cardiometabolic Phenotype"

_nutrients, 2025, doi:10.3390/nu17223614_

Round 1
Reviewer 1 Report
Comments and Suggestions for Authors
The manuscript presents an extensive and well-executed comparative analysis of two Western-style obesogenic diets—the cafeteria (CAF) and fast-food (FFD) diets on female Wistar rats over 12 months. The study combines behavioral, metabolic, and neuroinflammatory endpoints, offering valuable insights into how diet composition and food choice complexity shape obesity-associated phenotypes. The experimental design is comprehensive and the statistical analyses are largely appropriate. However, several mechanistic and methodological gaps reduce the overall impact and translational clarity.
Some of the narratives to be addressed
- The manuscript repeatedly attributes neuroinflammation and behavioral changes to microbiota-derived metabolites (e.g., indoxyl sulfate, 5-HIAA), yet no microbiome profiling was performed. This is a significant omission given the centrality of this mechanism in the Discussion.
- Rationale for using only female rats in this study, especially since diet-induced obesity and hypertension exhibit marked sex differences.
- The study assesses end-point parameters after 12 months but provides no longitudinal data. This makes it difficult to infer "cause effect relationships" between neuroinflammation and obesity onset.
- The reliance on GFAP and TNF-α as sole markers for neuroinflammation is insufficient to capture the complexity of glial activation.
- Reduced locomotion and social interaction are interpreted as anxiety-like or depressive behaviors. However, these could also reflect reduced activity due to obesity rather than altered affect.
- Figures use multiple comparison symbols (*, #, $) inconsistently and without explicit post-hoc explanations.
- The elevated body temperature in CAF-fed rats is intriguing but speculative without direct BAT activity or UCP-1 measurements.
- The authors acknowledge sex-dimorphic effects in diet-induced obesity and hypertension—citing male vs. female responses but do not link these to estrogen levels:
- Did not monitor estrous cycles or measure estrogen/estradiol
-
Did not perform ovariectomy to control for hormonal variation
-
Because estrogen exerts neuroprotective, anti-inflammatory, and metabolic effects, its omission limits interpretation of:
-
The severity of CAF-induced insulin resistance and neuroinflammation
-
The corticosterone elevation in FFD rats (potential HPA-axis/estrogen interaction)
-
The behavioral phenotypes (social withdrawal, anxiety-like behavior), which are known to vary with estrous stage
-
Reviewer 2 Report
Comments and Suggestions for Authors
This study was to compare cafeteria and fast-food diets with standard diets on the manifestation of obesity-related complications. The diets are western styled like around the world nowadays. It is interesting to read the results of the study. The description of the methods and results were in detail. Some comments as follows:
- The conclusion of the study might need to be reconsidered. The part about diet design was worth discussing. The group of the fast-food diet was served only one single type of chow, but the cafeteria diet group was offered four menus changed over two days along with a standard chow. The results of the two intervention groups were obviously different. Actually the settings of the two groups were not on the same page. The cafeteria diet group had more chances to obtain different kinds of food, while the fast-food diet had only one kind. Actually in the real world, fast-food could be present in different combinations of food. Food variety is related to food consumption. I don’t think cafeteria diet was really inferior to the fast-food diet regarding obesity development. In the current study, the differences of the two groups might result from the fast-food diet group not given as many food choices as the cafeteria diet group.
- The number of the references was excessive. It was normally under 40. The length of an article was better to be reduced.
Round 2
Reviewer 1 Report
Comments and Suggestions for Authors
Thank you for the thorough revision of "Cafeteria and Fast-food Diets Induce Neuroinflammation, Social Deficits, but a Different Cardiometabolic Phenotype". I’ve reviewed your responses and the updated manuscript, and I’m pleased to confirm that you have addressed all reviewer comments.